# Biologic Drugs for Rheumatoid Arthritis in the Context of Biosimilars, Genetics, Epigenetics and COVID-19 Treatment

**DOI:** 10.3390/cells10020323

**Published:** 2021-02-04

**Authors:** Krzysztof Bonek, Leszek Roszkowski, Magdalena Massalska, Wlodzimierz Maslinski, Marzena Ciechomska

**Affiliations:** 1Department of Rheumatology, National Institute of Geriatrics Rheumatology and Rehabilitation, 02-635 Warsaw, Poland; krzysztof.bonek@gmail.com (K.B.); leszekroszkowski@poczta.fm (L.R.); 2Department of Pathophysiology and Immunology, National Institute of Geriatrics Rheumatology and Rehabilitation, 02-635 Warsaw, Poland; magdalena.massalska@spartanska.pl (M.M.); wlodzimierz.maslinski@spartanska.pl (W.M.)

**Keywords:** bDMARDs, biologics, biosimilars, rheumatoid arthritis, genetics, epigenetics, COVID-19

## Abstract

Rheumatoid arthritis (RA) affects around 1.2% of the adult population. RA is one of the main reasons for work disability and premature retirement, thus substantially increasing social and economic burden. Biological disease-modifying antirheumatic drugs (bDMARDs) were shown to be an effective therapy especially in those rheumatoid arthritis (RA) patients, who did not adequately respond to conventional synthetic DMARD therapy. However, despite the proven efficacy, the high cost of the therapy resulted in limitation of the widespread use and unequal access to the care. The introduction of biosimilars, which are much cheaper relative to original drugs, may facilitate the achievement of the therapy by a much broader spectrum of patients. In this review we present the properties of original biologic agents based on cytokine-targeted (blockers of TNF, IL-6, IL-1, GM-CSF) and cell-targeted therapies (aimed to inhibit T cells and B cells properties) as well as biosimilars used in rheumatology. We also analyze the latest update of bDMARDs’ possible influence on DNA methylation, miRNA expression and histone modification in RA patients, what might be the important factors toward precise and personalized RA treatment. In addition, during the COVID-19 outbreak, we discuss the usage of biologicals in context of effective and safe COVID-19 treatment. Therefore, early diagnosing along with therapeutic intervention based on personalized drugs targeting disease-specific genes is still needed to relieve symptoms and to improve the quality of life of RA patients.

## 1. Introduction

Disease-modifying antirheumatic drugs (DMARDs) cover a group of otherwise unrelated drugs interfering with the disease process leading to rheumatoid arthritis (RA) [1]. The term DMARDs was used for many years by rheumatologists to distinguish them from nonsteroidal anti-inflammatory drugs (aimed to treat inflammation, without treating the underlying cause) and from steroids, which weaken the immune response but were not able to stop disease progression [1].

DMARDs have been classified as synthetic DMARDs (sDMARDs) and biological DMARDs (bDMARDs) [1]. Generally, biological drugs usually target extracellular or cell membrane molecules, while chemical compounds enter the cells and interact with intracellular structures [1]. The main function of bDMARDs is based on direct interaction with certain cytokine or surface molecule, which subsequently leads to their neutralization. In addition, bDMARDs can indirectly reduce the serum level of other cytokines in RA patients [2,3]. For instance, anti-TNF treatment resulted in the reduction of IL-8, IL-1 and MCP-1 which are present in inflamed synovium [4]. These studies suggest that bDMARDs have broad spectrum of action beyond their direct function due to complexity of proinflammatory network. sDMARDs hold conventional synthetic (csDMARDs) and targeted synthetic DMARDs (tsDMARDs). csDMARDs are the drugs (such as methotrexate, sulfasalazine, leflunomide, hydroxychloroquine, gold salts) that have entered the treatment proposals for RA in a conventional historic way, involving empiric and accidental findings of disease-modifying properties, without known target of the drug. On the contrary, tsDMARDs are the drugs that were developed to target – a particular, well-defined molecular structure (e.g., JAK inhibitors).

High costs of biologics are a main reason limiting patient access to these therapies. The high costs of TNF inhibitors (TNFi) have made them one of the most profitable drugs in the world. Indeed, adalimumab (ADA) Humira was sold for USD 18 billion in 2017 (being the world’s best-selling drug) and etanercept (ETN) Enbrel and infliximab (IFX) Remicade were sold for about USD 8 billion each [5]. Patent expiration of biologics has enabled development of lower-cost biosimilars, what opens on increased market competition and price reduction [6]. It is estimated that availability of biosimilars will result in reduction of direct costs spending on biologics by USD 54 billion in the U.S. from 2017 to 2026 [7]. UK National Health Service saved £ 324 million in the 2017/18 financial year by switching from using 10 expensive medicines, including IFX, to cheaper biosimilars [8]. Poor affordability followed with national reimbursement criteria, that can be more restrictive than the treatment guidelines, leads to disparities in patient access to biologics among countries [9]. Only 59% of patients eligible for biologics according to EULAR guidelines followed the therapy because of national reimbursement criteria in the European region [10].

Although the etiology of RA still remains elusive, the pathogenesis of RA is influenced by complex interactions between the immune system and genetic and epigenetic alterations [11]. Recent findings clearly implied the contribution of genetics influenced by epigenetic modifications like DNA methylation, histone modifications and presence of microRNA (miRNA) to disease susceptibility in RA [12]. The dominant RA risk alleles were class II major histocompatibility (*MHC*) but over 100 non-*MHC* risk alleles have also been identified including genetic variations in type I interferon-related genes [13]. Studies on identical twins have proved that environmental and behavioral factors (like smoke, diet and exposure to toxins or ultraviolet light) can result in epigenetic dysregulation, which influence the onset and severity of disease [14].

In this review we try to debate how personal genetic and epigenetic landscape modulates patient’s responses on bDMARDs administration. We also discussed the usage of biosimilars which are alternatives to already licensed bDMARDs therapies and reviewed available bDMARDs therapies as proposed in COVID-19 treatment.

## 2. bDMARDs Based on Cytokine-Targeted Therapy

### 2.1. TNF Inhibition

TNF is one of primary proinflammatory modulators in RA. TNF is able to stimulate NF-κB related response by activating INF-γ, IL-1, IL-6 pathways, resulting in bone erosions [15,16,17,18,19]. Currently, there are five classes of drugs targeting TNF pathway available in RA treatment: adalimumab, etanercept, certolizumab pegol, infliximab, and golimumab (Figure 1). TNFi are an important part of RA treatment strategy according to EULAR (2019 guidelines), ACR (2015 guidelines) and APLAR (2018 guidelines) [20,21,22]. According to ACR guidelines TNFi are recommended in patients with sustaining moderate and high disease activity measured using ACR approved indices: Clinical Disease Activity Index (CDAI), Disease Activity Score with 28-joint counts with erythrocyte sedimentation rate (DAS28-ESR) or C-reactive protein (DAS28-CRP), Patient Activity Scale (PAS), PAS-II, Routine Assessment of Patient Index Data with three measures, and Simplified Disease Activity Index (SDAI). Based on EULAR guidelines, TNFi are recommended in a second phase of treatment in patients with poor prognostic factors (high levels of ACPA and RF, early joint damage, high disease activity, failure of at least two csDMARDs) who fail to improve at least 50% over 3 months or not reaching remission or low disease activity at 6 months of therapy, preferably in dual therapy with methotrexate (MTX).

Combination therapy of TNFi and MTX has been proven superior over TNFi in monotherapy or MTX in monotherapy in terms of drug survivability, treatment efficacy (measured using DAS28, tender and swollen joint counts, CRP concertation), percentage of patients reaching EULAR’s good response criteria and overall patient outcomes [23]. All available TNFi share similar probability on reaching remission/low disease activity, response rate (30% primary nonresponders), cardiovascular risk reduction, and potential to stop radiographic progression, although recent data show that adalimumab seems to be most effective in geriatric patients, while etanercept is associated with lower risk of developing tuberculosis [24,25,26,27]. Importantly, from all bDMARDs only TNFi are approved during pregnancy and lactation. It has been demonstrated that prenatal exposition on TNFi does not influence development of T or B cells [28,29]. Yet, there are some safety concerns regarding risk of developing serious infections, such as tuberculosis due to detectable anti-TNF antibodies in infants sera [30]. According to the EULAR experts, some TNFi including certolizumab (CTZ) and ETN have been approved during pregnancy [31,32]. Treatment with infliximab and adalimumab can be continued up to 20 weeks, and with etanercept up to 30–32 weeks of pregnancy [32]. Certolizumab lacks an Fc receptor (FcR) part of antibody (Ab) what results in blocked transfer across the placenta, suggesting its safety during pregnancy [31,32,33]. Available data from two cohort studies, CRIB and CRADLE, led to its registration in EU for the treatment during pregnancy and breast-feeding periods [31,34,35]. Similarly, ETN due to its low concentrations in the breast milk and undetectable levels in newborn sera is considered by EULAR safe, although strength of evidence is lower in ETN than in CTZ [30]. Data on teratogenic effects of TNFi are limited and there is no strong evidence of potentially harmful effects, if used in preconception period [32]. Recent data and metanalysis suggest that biosimilar drugs are equal to their biooriginators, however, recent data have shown that treatment with biosimilar ADA, ETN and IFX might be associated with increased discontinuation rates due to medical and nonmedical issues, such as the nocebo effect [34,35,36,37,38,39]. Moreover, researchers from the UK suggest that nonmedical switch from biooriginal to biosimilar drug might involve higher costs from the medical system than continuation of the original drug [40].

#### 2.1.1. Etanercept (ETN)—TNF Receptor Fusion Protein

ETN was the first anti-TNF drug registered for treatment of RA [41]. ETN has a unique structure among TNFi. It is a soluble fusion protein consisting of two human TNF receptors and human Fc tail [41]. According to available research and recommendations, combination of etanercept and MTX have been found to improve radiographic and clinical outcomes over MTX naive patients, patients started with combined therapy (ETN + MTX) who switched to ETN monotherapy or ETN in monotherapy from the beginning of treatment [42,43]. Data acquired from PRESERVE trial have shown that in certain groups of patients combined therapy of ETN and MTX could lead to dose reduction or even drug discontinuation in patients in “deep remission” [43]. Introduction of ETN biosimilar led to revolution in RA treatment, for example in Denmark within 4 months almost 90% of patients treated with biologics were switched to biosimilars [44]. ETN biosimilar can be considered equivalent to biooriginator drug in management of RA as first line treatment [35]. Interestingly, ETN biosimilar has shown comparable potential to biooriginator to stop radiographic progression in RA [45]. Only proven similar efficacy and safety in short- and long-term trials allow switching between original drug and biosimilar [37,46,47]. However, such drug change was burdened with strong nocebo effect, thus lowering biosimilar’s efficacy [48,49].

#### 2.1.2. Infliximab (IFX)—A Humanized Mouse Monoclonal Antibody

IFX, under the brand name Remicade, was the first synthetized TNFi drug and it was registered in 1998 by FDA for treatment of RA. Treatment with IFX has similar probability of reaching remission/low disease activity, response rate, cardiovascular risk reduction, and potential to stop radiographic progressions as other TNFi [33,46]. Patients treated with IFX have shown the highest rate of treatment discontinuation in comparison to ETN and ADA groups [24]. Moreover, treatment with IFX was associated with higher rate of infections in comparison to golimumab (GOL), ETN, CTZ, and ADA [25]. Similarly, the first biosimilar drug approved by FDA in RA treatment was biosimilar IFX (Inflectra). In recent studies there were no significant differences between bioorignal and IFX biosimilar in terms of safety and efficacy [47]. In terms of immunogenicity of IFX biosimilar, no concerns were raised in comparison to other TNFi [50]. There were no significant clinical differences in terms of drug efficacy after a switch from biooriginator to IFX biosimilar in comparison to continuous biosimilar treatment [40,51,52]. However, in 49% of patients changing to IFX biosimilar the dose escalation was observed [53]. After 5 years of observation, there were no safety issues raised over patients treated solely with IFX biosimilars or biosimilars after switch from biooriginator drug, in comparison to patients treated only with original IFX [54]. Similar data were obtained from Danish DANBIO and NOR-SWITCH follow-up studies [55,56]. In addition, in 2019 EMA approved first subcutaneous IFX Remsina giving RA patients more control to treat themselves [51].

#### 2.1.3. Adalimumab (ADA)—Fully Human Monoclonal Antibody

Adalimumab is monoclonal antibody targeting soluble and transmembrane TNF. Similarly to other TNFis, it is used in second line treatment preferably with MTX along with other TNFis or IL-6 and JAK inhibitors [20,21,22]. There are currently available multiple ADA biosimilars. Several trials show that ADA biosimilars are comparable to bioriginator drugs in terms of efficacy, immunogenicity and safety in RA treatment [52,57,58]. There were no differences in ADA’s pharmacokinetics between biosimilar and bioorignator drug [59]. Moreover VOLTAIRE-RA study’s extension did not show any differences in effectiveness, nor side effects between ADA biosimilar and biooriginator in extended 2 years prospective observation after the end of original studies [59]. Another trial showed that there were no differences in immunogenicity, efficacy or safety even after one or two drug switches between biosimilar and biooryginator [60]. There was no increase in adverse events (AEs) in patients with ADA biosimilar and concomitant MTX therapy comparing to Humira and MTX treatment [61].

#### 2.1.4. Certolizumab (CTZ)—Humanized Antigen-Binding Fragment of a Monoclonal Antibody Binding TNF

Certolizumab is another biooriginator TNFi drug. Due to its lack of trans-placental transport, it has been approved in RA treatment during pregnancy [41,62]. Efficacy of CTZ seems to be comparable to other TNFi in RA treatment, although in the majority of patients neutralizing antibodies were found [63]. Yet, high concentrations of CTZ in sera (>10 mg/L) were able to sustain a sufficient clinical response in RA treatment [64]. Therefore, the presence of neutralizing antibodies could result from high concentrations of CTZ administration. In large cohort studies there were no significant safety issues in RA treatment [65], although in meta-analysis increased number of serious adverse events (SAE) and infections was observed in the CTZ group in comparison to other TNFi [63]. As of today, there are no CTZ biosimilar drugs present.

#### 2.1.5. Golimumab (GOL)—Fully Human Monoclonal Antibody

In comparison to other TNFi, GOL shows similar efficacy and safety [66,67], but seems to be less effective than other TNFis in patients with multiple biological treatment failures [68]. In comparative trials, there were no significant divergences in effectiveness or safety between GOL and other TNFis [69]. GOL also might be considered as a safe drug during lactation, because its high mass might prevent its exudation to breast milk [70]. There are no GOL biosimilars currently available.

### 2.2. IL-6 Inhibition

Numerous studies have shown the key role of pleiotropic cytokine IL-6 in the RA autoimmune network, contributing to the activation of B and T lymphocytes, the production of acute phase proteins, autoantibodies, and the stimulation of synoviocytes and osteoclasts. There are six drugs on the market focusing on neutralization of IL-6 (Figure 1). The first humanized monoclonal antibody against the IL-6 receptor (anti-IL-6R) is tocilizumab (TCZ), approved for the treatment of MTX or TNFi resistant RA patients. TCZ binds to both soluble and membrane bound IL-6R, preventing IL-6 from interacting with both IL-6R and the signal converter glycoprotein 130 complex [71]. This results in JAK-STAT pathway inhibition [72]. IL-6 together with other inflammatory mediators stimulate B cells and induce differentiation of T cells [73]. Consequently, IL-6 promotes the production of antibodies, causing B cell maturation and in combination with TGF-β induces naive T cells to differentiate into Th17 cells and increases the production of IL-17 [74]. Furthermore, IL-6 induces the secretion of positive acute-phase proteins by hepatocytes, mainly CRP and also initiates fibroblast-like synoviocytes (FLS), which are a significant source of cytokine secreted into the synovial fluid. IL-6 plays also an important role in modulating extraarticular manifestations of RA symptoms such as fatigue, anemia, bone loss, depression, type 2 diabetes, increased cardiovascular risk, and interstitial lung disease [75,76]. TCZ is available in subcutaneous (sc) and intravenous (iv) forms. The risk of immunogenicity of this drug is low [77]. A number of randomized controlled studies with TCZ have been conducted in various patient groups. For example, BREVACTA and SUMMACTA were carried out in patients who did not respond to bDMARDs or csDMARDs [78,79]. Patients, who were not treated with MTX before were examined in the FUNCTION and AMBITION studies [80,81]. The largest number of patients with an unsatisfactory response to csDMARDs were studied in LITHE, TOWARD and ACTION studies [82,83,84]. TCZ has been shown to be more effective than ADA alone and therefore appears to be the drug of choice for MTX intolerance (ADACTA study) [85]. Numerous experiences in clinical trials and real-world conditions in recent years have confirmed the efficacy of iv and sc TCZ in RA patients who have failed csDMARDs or bDMARDs. Safety of iv and sc TCZ was reported in 11 published phase 3 and 4 studies [78,79,80,81,82,83,84,85]. The most common AEs reported in RA patients treated with TCZ in these studies were infections such as nasopharyngitis, upper respiratory tract infections, pneumonia, and cellulitis. SAEs reported in RA patients treated with TCZ include gastrointestinal perforation, malignancies, myocardial infarction, and stroke. Laboratory abnormalities including decreased neutrophil counts, elevated liver enzymes and changes in lipid levels have also been reported.

Sarilumab is a human monoclonal antibody of the IgG1 subclass that acts selectively on IL-6R and has been approved for the treatment of RA patients who have an insufficient response to MTX or csDMARDs. This consent was based on the positive results of the phase 3, MOBILITY and TARGET clinical trials [86,87]. Very important in terms of sarilumab positioning in treatment hierarchy were the results of the randomized, double-blind, head-to head phase 3 study MONARCH, which showed that sarilumab was superior to ADA in terms of change from baseline in DAS28-ESR in RA patients and inadequate response to MTX [88].

Olokizumab is a humanized monoclonal antibody anti-IL-6. This antibody was tested for safety and efficacy in CREDO study in RA patients. The preliminary data show good efficacy, tolerability, and favorable safety profile of olokizumab [89]. Clazakizumab is a humanized anti-IL-6 monoclonal antibody, evaluated in a phase 2b study in patients with RA and insufficient response to MTX [90]. Vobarilizumab is a humanized monoclonal antibody blocking the IL-6 receptor. Its safety profile and efficacy in RA patients seems to be similar to TCZ [91]. Another IL-6 blocker, sirukumab was rejected by FDA in RA treatment due to concerns about occurrence of serious infections, malignancies and serious cardiac AEs [92]. There are no biosimilars currently available.

### 2.3. IL-1 Inhibition

Anakinra (ANK) is an antagonist of human IL-1R, which neutralizes the activity of IL-1α and IL-1β by competitively inhibiting binding to type I interleukin receptor. Based on experimental models, it has been hypothesized that IL-1β plays a major role in the destruction of bone and cartilage associated with RA. Additionally, plasma and synovial IL-1β levels in patients with RA correlate with disease activity and erosive disease [93,94]. ANK has been approved for the treatment of RA in combination with MTX in case of an insufficient response to MTX monotherapy. This drug is given as a daily sc injection. In RA therapy, ANK is less effective than TNFi, as demonstrated by meta-analysis [95]. It was also found that the risk of bacterial infections is higher than during MTX therapy [95]. ANK is not currently recommended for the treatment of RA. It is not included in the treatment algorithm according to EULAR recommendations of 2019 [20]. Global data from five clinical studies of ANK in RA showed that antagonists of human IL-1R are generally well tolerated and cumulative AEs did not increase significantly in patients taking ANK compared to placebo. A new meta-analysis of pooled data from 2771 RA patients confirmed the increased incidence of severe, dose-related infections [96]. Canakinumab is an anti-IL-1 monoclonal antibody that has been tested in a phase 2 clinical trial in RA patients with MTX failure [97]. In general, the results were better than those of ANK, but still worse than TNFi, thus canakinumab was not introduced in the treatment of RA, although approved by FDA in 2013 for juvenile idiopathic arthritis (JIA) treatment. The safety profile was also in line with previous experience with ANK, as infections were the most reported AEs.

Rilonacept is the IL-1R blocking drug which is a dimeric fusion protein consisting of the extracellular interleukin type 1 receptor domains and the IL-1R helper protein linked to the constant region (Fc) of human IgG1. Although it was hypothesized that rilonacept may be more clinically active than ANK, a phase 2 study in RA showed clinical efficacy was worse than TNFi and further clinical development of rilonacept in RA was suspended [98].

Clinical studies in recent years have evaluated many different drugs acting along with different cytokine pathways in RA. Unfortunately, most of them proved to be insufficiently effective in RA. Although inhibition of given cytokine was observed, IL-17A with secukinumab [99], ustekinumab (human monoclonal antibody targeting the IL-12/23 p40 subunit) and guselkumab (monoclonal antibody targeting IL-23 specifically), the treatment did not significantly reduce the signs and symptoms of RA [100]. The drugs targeting other cytokines including IL-7, IL-15, IL-18, IL-21, IL-32, and IL-33 are currently in clinical trials.

### 2.4. GM-CSF Inhibition

Granulocyte-monocyte colony-stimulating factor (GM-CSF) targeted therapies are another interesting group of drugs that are currently in clinical trials. The best known of them, mavrilimumab, is a monoclonal human IgG4 antibody with high affinity for the GM-CSFRα chain and low complement activation capacity due to the IgG4 Fc isotype. It is also a competitive GM-CSF antagonist [101]. In addition to its well-known hematopoietic role in the differentiation and proliferation of myeloid cells, GM-CSF is a proinflammatory cytokine that plays an important pathogenic role in autoimmune diseases such as RA. GM-CSF expression is induced by IL-12, IL-1α, IL-1β, and TNF, while it is inhibited by IL-10, IL-4 and IFN-γ. In an inflammatory environment, GM-CSF can recruit and activate resident and myeloid cells such as epithelial, endothelial, T-lymphocyte, and fibroblast populations [102]. For this reason, in refractory RA, blocking of the GM-CSF pathway by antibodies directed against the cytokine itself or its receptor has been studied (Figure 1). The effectiveness and safety of mavrilimumab has been confirmed in studies: the EARTH EXPLORER I and II [103,104]. The risk of infection observed with mavrilimumab appears to be lower compared to other csDMARDs and tsDMARDs. The overall safety profile of mavrilimumab appears to be very satisfactory, especially with regard to infection, although a long-term toxicity analysis is needed for a more comprehensive statement [105]. Another medicine in this group is gimsilumab—a human IgG1 monoclonal antibody against GM-CSF. Until now, the drug was evaluated only on the basis of phase 1 studies investigating safety and tolerability of the compound [106]. Another drug, otilimab (formerly known as MOR103), is a monoclonal recombinant human high affinity IgG1 antibody against GM-CSF. After promising results from phase 1 and phase 2 clinical trials in RA, the results of two phase 3 trials are awaited for completion [107]. Namilumab (human monoclonal IgG1 antibody that binds to the GM-CSF ligand with high affinity) and lenzilumab (recombinant IgG1κ monoclonal antibody, anti-GM-CSF) despite promising initial results, did not meet expectations in more advanced clinical trials. Phase 2 studies have started, but have been discontinued, and companies are not continuing with further research on these drugs [108].

## 3. bDMARDs Based on Cell-Targeted Therapy

### 3.1. T Cell Targeted Therapy

Targeting T cells has long been an aim for the treatment of RA and other rheumatic diseases, but the first attempts to target CD4^+^ T cells showed little clinical efficacy or significant AEs [109]. Abatacept (ABA) became the first bDMARD, which modulates T cell activation in RA. ABA is a recombinant fully human, soluble, fusion protein, and T cell costimulation modulator that consists of the extracellular domain of human cytotoxic T lymphocyte associated antigen-4 (CTLA-4), which is linked to the modified Fc region of the IgG1. Proper activation of T cells requires two signals from antigen presenting cells (APC): antigen recognition by the T cell receptor, and a second costimulatory signal such as CD80 and CD86 binding on the surface of the APC to the CD28 T cell receptor. Binding of ABA to the costimulatory molecules CD80 and CD86 on the surface of APC, blocks interaction with CD28 on T cells. In this manner autoreactive CD4^+^ T cells receive signal one in the absence of second signal, which leads to a state of T cell anergy or unresponsiveness. Thus, it prevents the production of cytokines and immune responses, that are the most important factors in the formation of inflammatory models of RA. Various studies have shown that ABA affects not only T cells but also other cells. ABA reduced the signs of polyclonal B cell activation in RA patients by lowering the levels of serum IgM, IgA, IgG, free light chains, and decreasing percentage of circulating post-switch memory B cells. Administration of ABA also reduced titers of rheumatoid factor (RF) and anti-citrullinated protein antibodies (ACPA), as well as levels of IL-6, soluble IL-2R, CRP, and soluble E-selectin and soluble ICAM-1 [110,111]. In vitro studies have shown that ABA prevented CD95-mediated apoptosis of both CD4^+^ T lymphocytes and regulatory T lymphocytes (Treg) [112]. ABA has been approved for the treatment of RA and is given to patients who have an insufficient response to one or more csDMARDs or TNFi [113]. ABA with MTX showed greater efficacy as compared to MTX in monotherapy which was shown in a survey of Kremer et al. [114]. It was found that treatment with ABA and MTX was associated with lower radiographic progression in RA patient than treatment with MTX alone [115]. ABA and MTX treatment were also associated with a significantly greater mean reduction in synovitis, osteomyelitis, and with significantly greater inhibition of bone erosion in RA patients [116]. In ACTION study, ABA has been effective in patients who have previously been unsuccessful in their treatment with a TNFi. However, data from a direct AMPLE study showed that sc ABA and MTX was noninferior to sc ADA and MTX [117]. The ATTEST study was unable to distinguish between ABA and infliximab in terms of tolerance, efficacy, and safety [118]. However, there was evidence of better response and better treatment efficacy when used ABA in early RA [119]. APIPPRA study launched to find a drug that could be effective in the preclinical phase of RA with the presence of disease-related serum autoantibodies [120]. This study will determine whether taking ABA at such an early stage of the disease is considered acceptable for a high-risk patient. ABA in combination with MTX was generally well tolerated in RA patients. In the pooled analysis of placebo-controlled studies, AEs were reported in 46.4% of placebo recipients and 51.8% of ABA recipients. The most common ABA side effects were mild headache, nausea, and upper respiratory tract infection. It is important that compared to other bDMARDs, ABA treatment in patients with RA was associated with an increased risk of melanoma. This was confirmed in an observational post-marketing study [121]. Interestingly it has been shown that bionaive patients had a longer survival to ABA as well as a better functional and clinical response to this drug compared to the bDMARD experienced patient [122].

### 3.2. B Cell Targeted Therapy

B cells targeted therapies are a currently rapidly developing therapeutic option in rheumatology. Indeed, according to PRAIRI study B cell affecting therapy has shown potential to reduce risk of development of RA from 40% to 10% in high-risk profile patients [123]. B cell therapies can be divided into several main therapeutic mechanisms. In particular, there are drugs leading to direct depletion of B cells peripheral pool (rituximab), indirect depletion via costimulation blockade (belimumab), plasma cell targeting (such as bortezomib), and B cell function inhibition (such as Bruton’s kinase inhibitors) [124,125,126]. As for today, the only drug registered for RA treatment is rituximab (RTX). RTX is a monoclonal chimeric mouse/human antibody targeting CD20 on B cells, therefore not affecting stem cells and plasmacytes [127]. Infusion of rituximab leads to fast depletion of mature B cells in lymphoid glands such as bone marrow but also in RA patients’ synovium [128]. RTX has been approved for treatment of RA by EULAR, ACR and NICE guidelines [21]. RTX is recommended in second or third line of treatment, in patients with sustaining moderate and high disease activity, preferably in dual therapy with MTX [20,21,129]. There are some studies suggesting that in patients failing to respond to two csDMARDs, RTX was noninferior to TNFi therapy [130]. Moreover, data acquired from SWITCH-RA study suggest that switching from TNFi to rituximab is associated with significantly improved clinical effectiveness compared to another TNFi treatment [131,132]. Meta-analysis of three clinical trials (DANCER, REFLEX and IMAGE) has demonstrated that RTX were comparable to placebo and MTX in terms of serious infections rate [96,133]. No increased risk of malignancies in patients treated for RA with rituximab was observed [134]. Yet, a major concern was raised on risk of reactivation of hepatitis B-virus (HBV) infection during RTX treatment compared to etanercept [133]. Similarly, in patients with anti-Hepatitis Virus B core Antibodies (anti-HBc) there was a risk of reactivation of HBV infection during RTX treatment compared to TNFi, anti-metabolites or glicocorticosteroids [129,130]. Since there is no data on RTX fetal toxicity in first trimester, the EULAR guidelines recommend stopping rituximab treatment 6 months before planning conception [32,135]. Due to the end of patent protections, RTX biosimilar was available since 2019. The studies have shown that during 52-week observation trial of RTX biosimilar infusion resulted in similar decrease of disease activity (CDAI, SDAI, DAS28) as biooriginator [136,137]. RA treatment with RTX biosimilar resulted in similar pharmacokinetic, efficacy and safety as biooriginator drug in up to 72 week long studies [136,137,138,139].

## 4. Biosimilars and Biomimics (or Intended Copies)

Introduction of biological treatment has changed modern treatment of RA. Although considered highly effective in treatment, social costs of maintaining biological treatment remains to be a serious challenge for social healthcare systems [140]. With expiring patents for biooriginator/reference drug, multiple medical manufacturers have begun development of their versions of existing drugs (Table 1). The developing process of biosimilar drugs is complicated and highly regulated. The biosimilar-developers’ first obstacle is manufacturing, using reverse engineering, highly “similar” and “comparable” products. Secondly, the biooriginators’ drug structure might change over time, as the manufacturer is allowed to implement changes to drug structure, for example to improve product stability or adjust to changing regulations [141]. The whole concept of “biosimilarity” is based on comparability with reference product in terms of structure, function, and biologic activity according to regional regulations based on the ‘totality of evidence’ for instance FDA, WHO, EMA, Japanese, and Australian regulations [142]. According to current knowledge, based on NOR-SWITCH trial, biosimilar drugs are considered noninferior to biooriginator drugs, with similar ratio of AE [143]. However, there were considerations regarding potential immunogenicity. Yet, current research on occurrence of monoclonal antibodies targeted against biological drugs show small differences between biosimilars and original products, but with no impact on immunogenicity [144]. According to EULAR and PARE statements, one switch from biooriginator to biosimilar drug can be considered, although it should not be due to financial reasons [145]. Moreover, starting the therapy with a biosimilar drug is considered equal to biooriginal [145]. Drugs that do not meet aforementioned regulations but are registered by local medical regulators are considered to be “biomimics” or “intended copies” [145,146]. There are several products available mostly in China, Peru and Mexico, although their use is controversial due to the lack of high-quality studies and safety issues [147]. Analyses from several batches of seven available etanercept-based biomimics has shown that none of them met minimum requirements proposed by WHO on biosimilarity [148]. RTX biomimic (Kikuzumab) was retracted from the market in Mexico due to a number of anaphylactic reactions during infusions, while another one (Reditux) failed to prove structural similarity to original drug [147]. Finally, Pan American League of Associations for Rheumatology (PANLAR) issued a statement that biomimics are not biosimilars and the use of biomimics is not recommended [149].

## 5. The Effect of Genetic Factors on bDMARDs Response in RA Treatment

It has been previously demonstrated that genetic factors including single nucleotide polymorphisms (SNPs) in proinflammatory cytokines such as TNF, IL-1β, and IL-6 or their downstream elements were associated with response to biologic therapy treatment in RA patients. For instance, it has been demonstrated that polymorphism in *IL-6* promoter region was correlated with better response to anti-TNF therapy [150,151]. Similarly, SNP in the promoter of *IL-32* helped to predict the response on ETN and ADA in RA patients [152]. On the other hand, the role of polymorphism in *TNF* remains controversial in RA patients undergoing TNFi therapy [153,154]. In addition, specific *IL-1RN* (variable number of tandem repeats) gene polymorphism was higher in the nonresponders group than in responders to TNFi therapy [155]. Polymorphisms within the steroid hormone related genes including *CYP3A4* and *CYP2C9* loci correlated with changes in DAS28 after treatment with anti-TNF drugs [156]. In addition, specific haplotype in *ESR2* gene was associated with a better response to anti-TNF therapy. Based on three large genome-wide association studies (GWAS) it has been discovered that 12 SNPs were associated with response to the TNFi [157]. These 12 SNPs were in the regions coding *CD84, CNTN5, NUBPL* among other genes. Surprisingly, SNP in *CD84* gene was associated only with the response to ETN, but not with the response to IFX or ADA. Our findings also demonstrated the potential associations between *NFκB1* polymorphisms and clinical outcome of biologic TNFi therapy in RA patients [158]. Indeed, patients carrying the *NFkB1* ins/ins genotype were characterized by worse response to TNFi treatment. Interestingly, some studies have reported that RA patients with specific *HLA* alleles can develop autoantibodies against TNFi [159,160]. These results suggest that specific *HLA* alleles play a role in formation of antidrug antibodies which results in decrease or even failure of biologic therapy in RA patients. Large GWAS study has demonstrated a strong association between polymorphism in *HLA-DBR1* gene and radiological damage in RA patients [161]. Furthermore, it has been observed that RA patients with *FcγRIIIA* gene polymorphism had better response to RTX, suggesting that the *FcγRIIIA* gene can modify the structure of therapeutic antibody binding [162,163]. Regarding ABA, no significant association was found between clinical response at 6 months and the SNPs in the *CTLA4, CD80,* and *CD86* genes [164]. So far there is a lack of information between polymorphism and biologic therapy using IL-6 inhibitors in RA. A recent meta-analysis in Chinese subjects identified new susceptibility loci in *IL12RB2, BOLL-PLCL1, CCR2, TCF7,* and *IQGAP1*. Interestingly, genes within these five loci are genetically associated with risk of RA. Furthermore, drug target enrichment analysis has found that encoding proteins of these genes can interact with currently investigated drug targets in RA, suggesting a possible usage of these findings in the clinics [165].

While large GWAS study revealed a few promising biomarkers based on SNPs which can predict positive outcome of bDMARDs therapy or radiographical damage in RA, however, none of these selected SNPs are directly used in the clinics so far. Therefore, more robust, and freely available data analysis should be accessible to combine and accelerate the effort to understand the RA pathogenesis in the context of precision medicine.

## 6. The Effect of Epigenetic Factors on bDMARDs Response in RA

Epigenetics is characterized by changes in gene expression that are not determined by changes in the DNA sequence and if dysregulated, it can result in the development of various pathological conditions, including RA. Indeed, epigenetic modifications including DNA methylation, miRNA expression and histone modification regulate gene expression, whereas altered epigenetic pattern may contribute to pathogenesis of RA [166,167].

DNA methylation is the most widely studied epigenetic mechanism, which consists of the addition of a methyl group at the cytosines followed by guanines (CpG dinucleotides). Such modification results in gene silencing. There are increasing studies focusing on the role of DNA methylation in RA FLS or peripheral blood mononuclear cells (PBMCs). Recently it has been demonstrated that changes in the monocyte methylome reflects disease activity in RA patients. Indeed, some CpG sites of *STAT3*, *FPR2*, and *TNFAIP8* correlate with DAS28 in RA monocytes [168]. However, the authors did not find any significant correlation between patient treatments (either biologic drugs or csDMARDs) and DNA methylation in RA monocytes. Other study demonstrated that RA FLS have distinct methylome and transcriptome profiles based on the joint origin [169]. Indeed, based on global analysis, RA FLS from the knee have hypomethylated genes related to IL-6 and JAK-STAT signaling compared to RA FLS from the hip. Thus, it raises the possibility that asynchronous responses might occur when IL-6 neutralizing antibodies or JAK-STAT inhibitors are used in RA therapies. In addition DNA methylation signature associated with cell migration, differentiation and adhesion pathways in early RA FLS is different compared to long standing RA FLS, suggesting that DNA methylation pattern occurs early and evolves over time [170].

MiRNAs, as epigenetic modulators, are involved in negative regulation of gene expression by degradation or inhibition of mRNA translation. Similarly to DNA methylation, altered miRNA expression also plays a role in RA pathogenesis and could be used as an attractive biomarker. Indeed, the expression level of miRNA-146a correlated with DAS28 in RA PBMCs [171]. In addition, microarray studies of circulating miRNA have shown that miRNA-146a is a good candidate predicting treatment outcome in RA patients undergoing anti-TNF therapy [172]. We have found that also SNPs in miRNA-146a sequences contribute to RA development. Furthermore, following 3 months of anti-TNF therapy, the level of miRNA-146a-5p increased, indicating that administration of biologic drugs gradually elevates the level of circulating miR-146-5p [158]. Similarly, we have also demonstrated that changes in miRNA-5196 expression in sera of RA and AS patients can be used as a biomarker predicting therapeutic response to TNFi therapy [173]. Surprisingly, based on receiver operating characteristics (ROC) analysis, changes in circulating miRNA-5196 expression were better predictors than changes inflammatory parameter CRP to anti-TNF therapy response in RA patients. Sode et al. have demonstrated that also miRNA-27a is a good candidate among the 91 validated miRNAs predicting the treatment response following 3 and 12 months of anti-TNF therapy [174]. Similarly, Krintel et al. found that combination of low miR-22 and high miR-886-3p was associated with a good response to ADA in early RA patients [175]. Recently it has been published that RA neutrophils treated with antibodies neutralizing TNF (IFX), but not neutralizing IL-6 (TCZ), increased the expression of miRNA-126, -29c, -30c, -17, -21, -223, let-7b [176]. These data suggest that IFX can restore the global levels of selected miRNAs and genes involved in neutrophils functions in RA. Therefore, dysregulation in miRNA expression could be used as a potential biomarker predicting the effective response to small molecules or biologic therapy in RA.

Histone modification also plays a role in RA pathogenesis. This epigenetic alteration is characterized by the addition of an acetyl, methyl, phosphorus, or other groups to histone proteins. Iraki et al. demonstrated that phosphorylation of STAT1, STAT3, and STAT5 in RA FLS was inhibited upon JAKi (peficitinib) treatment [177]. Studies conducted by Toussirot et al. have also demonstrated that histone acetyltransferase (HAT) activity increased considerably in RA and ankylosing spondylitis (AS) patients following anti-TNF (ETN, IFX or ADA) or RTX therapy treatment [178]. HATs are responsible for histone acetylation. Lin et al. have shown that TNFi (ETN and ADA) not only suppressed proinflammatory IL-17 and IL-22 but also downregulated the acetylation of histones H3 and H4 in the *RORγt* gene promoter region of Th17-polarized cells from RA patients, suggesting a close link between biologic therapy on histone modification [179].

Overall, these studies have shown that bDMARDs not only directly target inflammatory molecules but are able to indirectly change DNA methylation, miRNA expression, and status of histone modification in RA patients. Indeed, there are already a few clinical studies (NCT02742337, NCT02350491, NCT03984227) focusing on epigenetic signatures in RA and other autoimmune diseases treated with bDMARDs, which is the next step in translational medicine. Therefore, comprehensive genomic and epigenomic data collection about individuals is a crucial factor for effective bDMARDs treatment and overall precise RA management.

## 7. The Role of bDMARDs in Fighting COVID-19

Current research suggests that coronavirus disease 2019 (COVID-19) caused by severe acute respiratory syndrome coronavirus 2 (SARS-CoV-2) infection is associated with multiple phenotypes of disease progression due to massive release of proinflammatory mediators known as cytokine storm. The cytokine storm is mostly mediated by TNF, IL-1, IL-6, and interferons [180,181,182]. Therefore, blocking proinflammatory mediators (via inhibition of IL-1, IL-6, IFNs, TNF, GM-CSF), signaling pathways (JAK inhibitors) and lymphocytes functions (anti-CD20) may reduce cytokine storm and potentially save patients from developing acute respiratory distress syndrome (ARDS) or severe organ damage [183]. Since bDMARDs can directly neutralize certain cytokines and subsequently reduce proinflammatory network in RA, therefore those drugs have been proposed as a treatment option for COVID-19 where cytokine storm is one of the main factors of COVID-19 pathogenesis.

Due to promising observations made on small groups of patients with pulmonary involvement, there is currently an undergoing trial of ADA in SARS-CoV-2 infection (ChiCTR2000030089). Another anti-COVID-19 strategy is IL-6 blockade. Indeed, the symptoms like hypoxemia and computed tomography (CT) opacity changes were improved after the treatment with tocilizumab in most of the patients, suggesting that TCZ could be efficient therapeutically [184]. In COVID-19 patients associated with severe pneumonia, it was found that the use of TCZ reduced risk of mechanical ventilation and subsequently the patient outcome was improved despite severe infections and AEs enhancement [185,186,187]. On the other hand, in patients which are already mechanically ventilated, administration of TCZ did not improve the patient’s outcome [187]. The outcome of another trial, the REMDACTA study (TCZ and remdesivir vs. placebo) should be available in early 2021 [188]. However, recently published case report demonstrated that treatment with TCZ was associated with potential risk of hypertriglyceridemia and acute pancreatitis in COVID-19 patients [189]. Additionally, a study from several Boston hospitals showed that TCZ was not effective in preventing intubation or death in hospitalized COVID-19 patients [190]. According to COVID-19 treatment guidelines published by NIH in November 2020 the usage of bDMARDs is recommended to treat COVID-19 patients only under the supervision of clinical trials [191]. One of the most recent published studies found that in COVID-19 patients with pneumonia, who were not receiving mechanical ventilation, TCZ reduced the probability of progression or death, but did not improve survival [192]. Very satisfactory results were obtained in the preliminary results of the REMAP-CAP study in which the IL-6 inhibitors (TCZ and sarilumab) were administered. It was found that both drugs can improve the survival rate of patients with severe COVID-19, as well as reduce the need for intensive care. The combination of IL-6R antagonists and corticosteroids has been shown to be of greatest benefit [193]. Other studies using sarilumab are also ongoing (NCT04315298). ANK is also considered in COVID-19 treatment due to its safety and wide therapeutic spectrum in controlling cytokine storm [194]. The effectiveness and safety of anakinra in patients with severe COVID-19 infection is currently under an investigation (NCT04603742). In addition, due to the high level of GM-CSF in COVID-19 patients, the research blocking GM-CSF has begun [182]. As for JAKi, there are ongoing 11 trials with ruxolitinib and seven with baricitinib according to records registered on ClincalTrials.gov. Importantly, the use of RTX in COVID-19 therapy needs more attention. It has been demonstrated that patients with granulomatosis with polyangiitis infected with SARS-CoV-2 and treated with RTX developed rapid deterioration, delayed recovery and worse outcome after severe COVID-19 pneumonia [195,196]. Additionally, there are reports of two RA patients who died due to severe infection COVID-19 during RTX treatment [197]. Aggressive course of COVID-19 in immunodeficient patients might result from the fact that they have been severely immunocompromised by the B cells depletion and application of glucocrticosteroids. Noteworthy, death of another two COVID-19 patients on RTX due to persistent viremia and subsequent pneumonia suggests that careful monitoring is recommended during RTX administration [198].

Another problem to consider is increased infectious risk, including SARS-CoV-2 in rheumatic patients. This risk seems to result mostly from the autoimmune disease itself but is also related to iatrogenic effect of immunosuppressive agents such as corticosteroids and csDMARDs or bDMARDs. Moreover, patients treated with TNFi were at lower risk of poor outcomes of COVID-19 infection in comparison to patients taking steroids [199]. Interestingly, although RA patients on higher doses of bDMARDs have almost 2-fold increased risk of serious infections compared to csDMARDs, they were at reduced risk of COVID-19 upon ts/bDMARDs administration [200,201,202]. Furthermore, patients with chronic arthritis treated with DMARDs did not seem to be at higher risk of life-threatening complications than the general population [203]. According to EULAR and ACR guidelines, RA patients should not stop or reduce their treatment during COVID-19 [204]. Furthermore, RA patients, in whom even brief drug holidays would be expected to cause flare of their disease should be supplied by sufficient IL-1 and IL-6 and JAK antagonists [205].

## 8. Conclusions

The use of bDMARDs has revolutionized the treatment of RA. It is well known that early and aggressive intervention using biologics in RA patients, who have an inadequate response to csDMARDs, results in significant reduction of disease progression. In this review we highlighted the recent advantages and side effects of the bDMARDs, which are on the market or in the development stage including blockers of TNF, IL-6R, IL-1, GM-CSF, T cells, and B cells. Importantly, the patents of five TNFi and B cells blockers are either expired or will soon expire, therefore we also summarized the challenges faced by using biosimilars. Indeed, administration of biosimilars offers several potential advantages including more treatment options, increasing the access to lifesaving medications, and lowering the health care costs through competition with original drugs. Subsequently, these mechanisms result in treatment of additional RA patients. In addition, it has been suggested that genetic and epigenetic background play a role in response to bDMARDs therapy. Therefore, understanding the genetic and epigenetic mechanisms allows to decipher individuals’ unique molecular signatures and to stratify patients for more efficient and personalized bDMARDs treatment. Finally, we also highlighted promising results of bDMARDs administration in fighting COVID-19. Due to the common inflammatory-mediated pathogenesis of COVID-19 and autoimmune diseases, the use of biologic drugs targeting specific inflammatory proteins is considered as a possible treatment option. However, larger-scale studies are necessary before affirming that biologics do not expose COVID-19 patients to an increased risk of complications.

## Figures and Tables

**Figure 1 cells-10-00323-f001:**
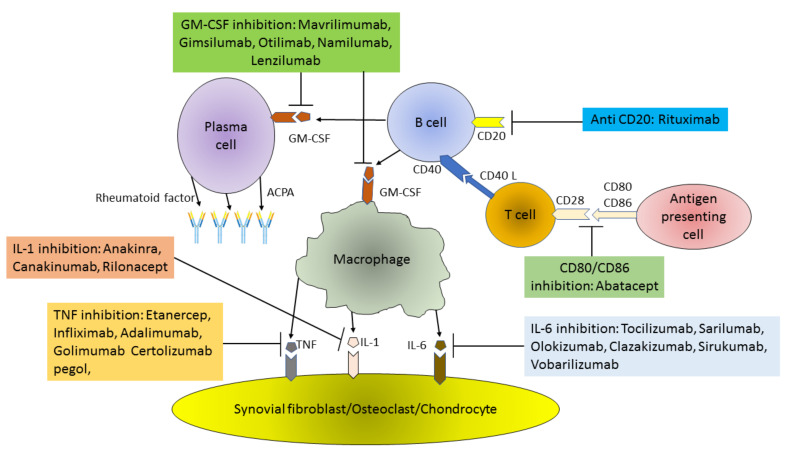
Pathogenesis of RA (Rheumatoid arthritis) and main targets of biological therapy. ACPA – Anti CCP Ab: Anti-Citrullinated Protein Antibodies; GM-CSF: Granulocyte-Macrophage Colony Stimulating Factor; TNF: Tumor Necrosis Factor; IL: Interleukin.

**Table 1 cells-10-00323-t001:** Originator bDMARDs, their current development stage in COVID-19 and appropriate biosimilars.

Drug Class	Name (Year Approved)	Current Development in COVID-19	Biosimilars
TNF inhibition	Etanercept–Enbrel (1998)		Erelzi (2016 `) Benapali (2016 *) Eticovo (2019 ^)
	Infliximab-Remicade (1999)	Phase 2 NCT04425538	Remisima (2013 *) Inflectra (2016 `) Flixabi (2016 *) Renflexis (2017 ^) Ixifi (2017 ^) Zessly (2018 *) Avsola (2019 ^)
	Adalimumab-Humira (2002)	Phase 2 ISRCTN33260034 Phase 3 IRCT20171105037262N4 Phase 4 ChiCTR2000030089	Amgevita (2016 `) Cyltezo (2017 `) Imraldi (2017 *) Solymbic (2017 *) Halimatoz (2018 *) Hefiya (2018 *) Hyrimoz (2018 `) Idacio (2018 *) Kromeya (2019 *) Hadlima (2019 ^) Abrilada (2019 ^) Hulio (2020 `)
	Certolizumab pegol-Cimzia (2009)		
	Golimumab-Simponi (2009)		
IL-6 inhibition	Tocilizumab-Actemra/RoActemra (2010)	Selected Phase 2 NCT04445272, NCT04479358, NCT04317092, NCT04331795, NCT04332094, NCT04377659, NCT04412291, NCT04479358, NCT04317092, NCT04435717, NCT04331795 Selected Phase 3 NCT04345445, NCT04412772, NCT04345445 Phase 4 NCT04377750, NCT02735707	
	Sarilumab-Kevzara (2017)	Selected Phase 2 NCT04357808, NCT04359901, NCT04315298, NCT04357860, NCT04324073 Phase 3 NCT04315298, NCT04327388, NCT04324073 Phase 4 NCT02735707	
	Olokizumab (phase 3)	Phase 3 NCT04452474, NCT04380519	
	Clazakizumab (phase 2b)	Phase 2 NCT04381052, NCT04348500, NCT04343989, NCT04363502, NCT04343989 Phase 3 NCT04351724	
	Vobarilizumab (phase 3)		
	Sirukumab (phase 3)	Phase 2 NCT04380961	
IL-1 inhibition	Anakinra-Kineret (2001)	Selected Phase 2 NCT04366232, NCT04462757, NCT04412291, NCT04443881, NCT04357366 Selected Phase 3NCT04424056, NCT04364009, NCT04443881, NCT04362111 Phase 4 NCT02735707	
	Canakinumab (phase 2)	Phase 2 NCT04365153 Phase 3 NCT04362813, NCT04510493	
	Rilonacept (phase 2)		
GM-CSF inhibition	Mavrilimumab (phase 2b)	Phase 2 NCT04463004, NCT04399980, NCT04397497, NCT04447469 Phase 3 NCT04447469	
	Gimsilumab (phase 1)	Phase 2 NCT04351243	
	Otilimab (phase 3)	Phase 2 NCT04376684,PER-042-20 EUCTR2020-001759-42-GB	
	Namilumab (phase 2)		
	Lenzilumab (phase 2)	Phase 3 NCT04351152, NCT04534725	
T-cell targeted therapy	Abatacept-Orencia (2005)	Phase 2 NCT04477642, NCT04472494	
B cell targeted therapy	Rituximab-MabThera/Rituxan (2006)		Truxima (2018 `) Ruxience (2019 `)

^—only FDA approval; `—FDA and EMA approval; *—only EMA approval.

## Data Availability

Not applicable.

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
