# Peer review of "Biologic Drugs for Rheumatoid Arthritis in the Context of Biosimilars, Genetics, Epigenetics and COVID-19 Treatment"

_cells, 2021, doi:10.3390/cells10020323_

Round 1

Reviewer 1 Report

-The title is not accurate : what are the « new perspectives » here ?

-The purpose of this paper needs to be better defined and focused, for example : original biologics and biosimilars in rheumatoid arthritis and severity of Covid-19 in RA patients treated with biologics. If the purpose is RA, the review should not integrate Covid-19 treatment in the general population.

-The abstract and the introduction should focus on RA and not RMD or rheumatology.

-As IL-6, IL-1 or GM-CSF effects have been largely developped in this review, TNF role may be detailed a little bit more.

-The second line of treatment in patients with sustaining moderate and high disease activity needs to be explained.

-Adalimumab also targets transmembrane TNF.

-Infliximab should be set before adalimumab.

-Data with subcutaneous infliximab may be added.

-Some similar data with TNF may be pooled to limit redundancy.

-Line 175 : DANBIO registry

-Certolizumab pegol is not a new biooriginator anymore.

-Differences in efficacy and drug survival with mono vs combo therapy may be detailed.

-Pregnancy should be detailed with all biologics and not only with Tnf inhibitors. Similarly to pregnancy, lactation with TNF inhibitors may be detailed.

-As the authors phave presented APIPPRA, they may present PRAIRI with rituximab.

-Table 1 : Amgevita

-mIRs do not negatively regulate gene expression but regulate genes by degrading or inhibiting mRNA translation.

-Fig 1 : anti-CCP should be replaced by ACPA. Self-explanatory caption should be added. GM-CSF receptor is mainly expressed by myeloid cells and not plasma cells.

Author Response

-The title is not accurate : what are the « new perspectives » here ?

-The purpose of this paper needs to be better defined and focused, for example : original biologics and biosimilars in rheumatoid arthritis and severity of Covid-19 in RA patients treated with biologics. If the purpose is RA, the review should not integrate Covid-19 treatment in the general population.

Thank you for this comment. We have removed “new perspective” from the title. Since the reviewer no 3 was asking to expand the genetic and epigenetic section, which we did, we decided to keep genetics and epigenetics in the title.

-The abstract and the introduction should focus on RA and not RMD or rheumatology.

This comment has been addressed and RDM was removed from the introduction.

-As IL-6, IL-1 or GM-CSF effects have been largely developped in this review, TNF role may be detailed a little bit more.

Thank you for this comment. Additional sentences regarding TNFi have been included. Lines 91-105

-The second line of treatment in patients with sustaining moderate and high disease activity needs to be explained.

This has been addressed. Lines 91-101

-Adalimumab also targets transmembrane TNF.

This has been addressed. Line 174

-Infliximab should be set before adalimumab.

This has been addressed.

-Data with subcutaneous infliximab may be added.

This has been addressed. Lines 171-172

-Some similar data with TNF may be pooled to limit redundancy.

Similar data from IFX, ADA and ETA sections have been removed.

-Line 175 : DANBIO registry

This has been corrected. Line 170

-Certolizumab pegol is not a new biooriginator anymore.

This has been corrected.

-Differences in efficacy and drug survival with mono vs combo therapy may be detailed.

This has been addressed. Lines 102-105

-Pregnancy should be detailed with all biologics and not only with Tnf inhibitors. Similarly to pregnancy, lactation with TNF inhibitors may be detailed.

The role TNFi during pregnancy and lactation has been expanded. Lines 110-114, 121-122

-As the authors phave presented APIPPRA, they may present PRAIRI with rituximab.

This has been addressed. Line 374-375

-Table 1 : Amgevita

This has been corrected.

-mIRs do not negatively regulate gene expression but regulate genes by degrading or inhibiting mRNA translation.

This has been corrected. Line 507-508

-Fig 1 : anti-CCP should be replaced by ACPA. Self-explanatory caption should be added. GM-CSF receptor is mainly expressed by myeloid cells and not plasma cells.

This has been corrected.

Reviewer 2 Report

The authors present an overview of bDMARDs and their biosimilars in the field of rheumatology. They nicely describe all possible treatment options with bDMARDS. I have only a few minor suggestions for improvement.

  • On page 4 line 139 (paragraph about etanercept) it is indicated that researchers from the UK suggest that a switch to a biosimilar might involve higher costs. It might be informative to elute a bit on this and explain their findings. This is only mentioned or etanercept was this the only drug that was assessed in this way or is this not a problem for the other drugs ?
  • In line 175 The Danish registry is called DAINIBO, this is a typo, it should be DANBIO.
  • In the paragraph about the genetic factors I miss some information to assess how we might translate (or not) these findings to the clinic. Some candidate genes are described but these are sometimes from relatively old studies and no replicated. In general replication in large (clinical) studies is lacking thus it is still the question if the findings are of clinical use. Still these studies can be very valuable to investigate the mechanisms of action. But this might require larger (homogeneous) cohorts. The same critical evaluation I miss for the epigenetic factors.

Author Response

The authors present an overview of bDMARDs and their biosimilars in the field of rheumatology. They nicely describe all possible treatment options with bDMARDS. I have only a few minor suggestions for improvement.

On page 4 line 139 (paragraph about etanercept) it is indicated that researchers from the UK suggest that a switch to a biosimilar might involve higher costs. It might be informative to elute a bit on this and explain their findings. This is only mentioned or etanercept was this the only drug that was assessed in this way or is this not a problem for the other drugs ?

This has been addressed. Line 125-130

In line 175 The Danish registry is called DAINIBO, this is a typo, it should be DANBIO.

This has been corrected. Line 170

In the paragraph about the genetic factors I miss some information to assess how we might translate (or not) these findings to the clinic. Some candidate genes are described but these are sometimes from relatively old studies and no replicated. In general replication in large (clinical) studies is lacking thus it is still the question if the findings are of clinical use. Still these studies can be very valuable to investigate the mechanisms of action. But this might require larger (homogeneous) cohorts. The same critical evaluation I miss for the epigenetic factors.

The translational aspect of genetic and epigenetic factors in the clinical settings has been addresses. Lines 465-466, 472-482, 545-547.

Reviewer 3 Report

This is a narrative review about biologic DMARDs in rheumatoid arthritis in the context of covid-19 pandemic.

Overall, this review is comprehensive, well written and the references are appropriate. My suggestion to the Authors is to focus on a single aim.

The stated aims are to provide a new perspective on biologic drugs and genetic/epigenetic mechanisms; to discuss biosimilars; to describe DMARDs and covid-19. These objectives have little relationship to each other.

I would suggest the Authors to shorten the paragraphs about DMARDs and biosimilars and make them introductory, if they wish to retain the important message about genetic and epigenetic factors related to bDMARD therapy.

The section on bDMARDs and covid-19 is interesting (although there are several reviews on the same topic) and may be retained; but it should also be improved ( e.g. extending the paragraph about the potential mechanisms of action of biologic drugs in covid-19; add more recent concluded trials).

Minor comments:

- I would number the paragraphs consecutively (e.g. 1. Introduction, etc).

- Table 1, Legend: "originator" instead of "original"

- English is fine but minor spell checks are needed (e.g. page 2, line 78, alternations instead of alterations).

Author Response

This is a narrative review about biologic DMARDs in rheumatoid arthritis in the context of covid-19 pandemic.

Overall, this review is comprehensive, well written and the references are appropriate. My suggestion to the Authors is to focus on a single aim.

The stated aims are to provide a new perspective on biologic drugs and genetic/epigenetic mechanisms; to discuss biosimilars; to describe DMARDs and covid-19. These objectives have little relationship to each other.

I would suggest the Authors to shorten the paragraphs about DMARDs and biosimilars and make them introductory, if they wish to retain the important message about genetic and epigenetic factors related to bDMARD therapy.

We have slightly expanded the bDMARDs section regarding role of TNFi during pregnancy and lactation, mono vs combo therapy and other questions, which were asked by reviewer no 1, however similar data from IFX, ADA and ETA sections have been removed to shorten the paragraphs about DMARDs. In addition, the section regarding genetic and epigenetic factors in bDAMRDs therapy in RA has been expanded. 465-466, 472-482, 545-547.

The section on bDMARDs and covid-19 is interesting (although there are several reviews on the same topic) and may be retained; but it should also be improved ( e.g. extending the paragraph about the potential mechanisms of action of biologic drugs in covid-19; add more recent concluded trials).

We have added a few recent findings regarding bDAMRDs in COVID-19. Lines 573, 577-587, 589-590. In addition, the paragraph about the potential mechanisms of action of biologic drugs in covid-19 based on their mechanisms of action of in RA has been extended. Lines 42-47, 559-562.

Minor comments:

  • - I would number the paragraphs consecutively (e.g. 1. Introduction, etc).
  • - Table 1, Legend: "originator" instead of "original"
  • - English is fine but minor spell checks are needed (e.g. page 2, line 78, alternations instead of alterations).

These comments have been amended.

Round 2

Reviewer 3 Report

The Authors correctly addressed the suggested changes. I have no further comments.

English is fine although some minor spell checks are still needed.